# 3D Fast Object Detection Based on Discriminant Images and Dynamic Distance Threshold Clustering

**DOI:** 10.3390/s20247221

**Published:** 2020-12-17

**Authors:** Baifan Chen, Hong Chen, Dian Yuan, Lingli Yu

**Affiliations:** 1School of Automation, Central South University, Changsha 410083, China; chenbaifan@csu.edu.cn (B.C.); hong.c@csu.edu.cn (H.C.); dianyuan@csu.edu.cn (D.Y.); 2Hubei Key Laboratory of Intelligent Robot, Wuhan Institute of Technology, Wuhan 430073, China

**Keywords:** vehicle-mounted lidar, discriminant image, region of interest, dynamic distance threshold clustering

## Abstract

The object detection algorithm based on vehicle-mounted lidar is a key component of the perception system on autonomous vehicles. It can provide high-precision and highly robust obstacle information for the safe driving of autonomous vehicles. However, most algorithms are often based on a large amount of point cloud data, which makes real-time detection difficult. To solve this problem, this paper proposes a 3D fast object detection method based on three main steps: First, the ground segmentation by discriminant image (GSDI) method is used to convert point cloud data into discriminant images for ground points segmentation, which avoids the direct computing of the point cloud data and improves the efficiency of ground points segmentation. Second, the image detector is used to generate the region of interest of the three-dimensional object, which effectively narrows the search range. Finally, the dynamic distance threshold clustering (DDTC) method is designed for different density of the point cloud data, which improves the detection effect of long-distance objects and avoids the over-segmentation phenomenon generated by the traditional algorithm. Experiments have showed that this algorithm can meet the real-time requirements of autonomous driving while maintaining high accuracy.

## 1. Introduction

Real-time and accurate object detection is essential for the safe driving of autonomous vehicles. If detection is lagging or inaccurate, autonomous vehicles may issue wrong control instructions and cause serious accidents. At present, the object detection algorithm based on images [1,2] is becoming more mature, and there are many improved algorithms based on these algorithms [3,4]. According to the benchmark report of KITTI [5], the average accuracy of the latest object detection algorithm based on images can achieve an average accuracy of about 80%. However, due to the lack of information related to the objects’ depth, object detection based on images is not enough to provide sufficient information for autonomous vehicles to perform planning and controlling. Although we can restore the spatial structure of the object through a matching algorithm, such as Mono3D [6], Deep3DBox [7], and 3DOP [8], the calculation amount is large, and the depth information recovered from them has errors. Therefore, direct 3D-data input is necessary. As the cost of lidar decreases, object detection based on point cloud data will be more widely used, and it will be able to provide high-precision, highly robust obstacle information for the safe driving of autonomous vehicles.

Point cloud is a set of points with a large amount of data that is different from the image, as it can reach hundreds of thousands of points in a single frame and contains a large number of ground points. It brings great challenges to object detection.

In the early days, most point cloud object detection methods were based on the traditional point cloud processing method, and they can be divided into three categories: object detection methods based on mathematical morphology, object detection methods based on image processing, and feature-based object detection methods. The object detection method based on mathematical morphology mainly performs morphological object detection on point cloud data; Linden-berger [9] first adopted this method in lidar object detection by using opening operation, which is a filtering method based on mathematical morphology, to process point cloud data, and then using auto-regression to improve the results. However, the method has limitations, since point cloud data are irregular, discrete spatial points. The object detection method based on image processing mainly converts the lidar point cloud data into a range images, and then uses image-processing algorithms for object detection. For example, Stiene et al. [10] proposed a CSS (Eigen-Curvature Scale Space) feature extraction method. This method extracts the contours and silhouettes of objects in range images and implements object detection using supervised learning. The basic idea of the feature-based object detection algorithm is to first obtain features, such as object height, curvature, edge, and shadow, then conduct conditional screening, and finally use clustering or recombination methods to obtain suspected targets. Yao et al. [11] used adaptive algorithms to obtain local features of objects, and constructed vehicle classifiers for object detection. Ioanou et al. [12] divided the scene point cloud based on the vector difference to obtain multiple interconnected spatial point cloud clusters, and then extracted the objects through clustering. In addition, there are other object extraction methods based on machine learning such as k-means and DBSCAN [13]. By selecting appropriate clustering parameters, the detection target is obtained in an unsupervised learning manner, but the parameter selection is difficult, it is easy to cause the problems of under-segmentation and over-segmentation, and it is not good for clustering sparse point clouds at long distances.

Several recent methods have adopted the deep learning method for point cloud object detection. According to different data-processing forms, it can be divided into three categories: object detection based on voxel grid, object detection based on point cloud projection, and object detection based on original points. The object detection method based on voxel grid divides the point cloud space into small cubes, called voxels, uniformly, which are used as the index structure unit in 3D space. For example, Zhou et al. [14] proposed VoxelNet, which is an end-to-end point cloud target detection network. It uses stacked voxel feature encoding (VFE) for feature learning, and extracts 3D features of the region proposal network for three-dimensional(3D) target detection. However, the calculation efficiency of this method is not high, and it cannot meet the real-time requirements of autonomous vehicles. The object detection method based on point cloud projection projects point cloud data in a certain direction, and then uses a deep neural network to perform object detection based on the projected image before, inversely transforming back to get the bounding box of the 3D object. For example, BirdNet [15] and RT3D [16] generated suggestion frames of object detection in 3D space from a bird’s eye view, but the results were not good. LMNet [17] takes the front view as input, but due to the loss of details, even for simple tasks such as car detection, it cannot obtain satisfactory results. Although VeloFCN [18] can accurately obtain the detection box of the object, the algorithm runs very slowly, and it is difficult to meet the real-time requirements of autonomous driving. The object detection method based on the original point cloud directly operating on the original point cloud data without converting the point cloud data into other data formats. The earliest realization of this idea was PointNet [19], which designed a lightweight network T-Net to solve the rotation problem of point cloud and used maximum pooling to solve the disorder problem of point cloud. On this basis, PointNet++ [20] performs local division and local feature extraction on point cloud data, enhancing the generalization ability of the algorithm. Although the semantic segmentation information can be obtained, the task it completes is the classification of the points, and the detection frame of the object is not obtained.

Usually, point cloud object detection methods first preprocess the original data, such as through ground segmentation and downsampling, and then cluster the acquired non-ground points to detect the objects. However, existing algorithms basically operate directly on the original data, so it is difficult to meet real-time requirements due to the great amount of original point cloud data. If the point cloud data are downsampled to reduce the number of scan points processed during calculation, part of the point cloud data is discarded, affecting the data integrity of the target point cloud. Moreover, when the object to be detected is far away from the lidar, the point cloud of the target surface becomes sparse, and the traditional Euclidean clustering method [13] for object detection can easily cause the problem of over-segmentation of objects at a long distance.

Based on these problems, this paper proposes a fast object detection algorithm based on vehicle-mounted lidar. The algorithm includes three modules: Module 1 uses the ground segmentation by discriminant image (GSDI) method to realize ground segmentation. It converts the original point cloud data into a discriminant image firstly and then uses breadth-first search (BFS) to traverse discriminate images to judge whether a point is ground point. It avoids direct point cloud data computing. Module 2 first uses the mature detector to obtain the object’s 2D detection boxes and then projects the object’s 2D detection boxes to 3D point cloud. Thus, the interest areas in 3D space can be obtained, and this can improve the search efficiency and ensure the integrity of the objects. Considering that the difference of the point cloud distance will cause the density of the point cloud to be different. Module 3 adopts a DDTC method to detect objects, it uses an adjustable parameter to determine the distance threshold when clustering points at different distances. Compared with the traditional Euclidean clustering method, it effectively solves the problem of over-segmentation of long-distance objects. Figure 1 shows the framework of the entire algorithm. Experiments have verified that the algorithm proposed in this paper has a good detection effect in different scenarios, and it can meet the real-time requirements of autonomous driving while maintaining high accuracy.

## 2. Three-Dimensional Fast Object Detection Algorithm

This section describes the three parts of our method (detailed in Section 2.1, Section 2.2 and Section 2.3) that correspond to ground segmentation, dynamic distance threshold clustering, and region of interest, respectively.

### 2.1. Ground Segmentation

The ground segmentation in the point cloud is the basis of tasks such as detection, recognition, and classification of obstacles. The ground segmentation method based on occupied grid map [21] is a commonly used method; this method converts point cloud data into a grid map, and the calculation speed of the algorithm is improved. However, it has a serious misdetection for the suspended planes. Another common algorithm for ground segmentation is based on ground plane fitting [22]. Unlike the ground segmentation method based on occupied grid map, it operates directly on the original point cloud data, without causing data loss. However, the calculations are huge and cannot meet the real-time requirements of autonomous driving. This paper proposes a GSDI method which first converts the original point cloud data into lidar images and then generates a discriminant image based on lidar images, to realize segment ground points quickly.

It can be known from the scanning characteristics of the lidar that the return value of the scanning point not only includes the distance, d, from the point to the lidar, but also includes the rotation angle, α, and the pitch angle, β. Different from the method of projecting a point cloud by occupancy grid map, this paper defines a new type of point cloud organization, which is called the lidar image. The conversion of the lidar image to 3D points can be achieved according to the following formulas:(1)$$x_{i}=d_{i}cos\left(\alpha_{i}\right)cos\left(\beta_{i}\right)=d\left(u,v\right)cos\left(v\delta_{v}\right)cos\left(u\delta_{u}\right);$$
(2)$$y_{i}=d_{i}cos\left(\alpha_{i}\right)sin\left(\beta_{i}\right)=d\left(u,v\right)cos\left(v\delta_{v}\right)sin\left(u\delta_{u}\right);$$
(3)$$z_{i}=d_{i}sin\left(\beta_{i}\right)=d\left(u,v\right)sin\left(v\delta_{v}\right).$$

In these equations, d(u,v) refers to the gray value of the coordinate (u,v) in the lidar image, indicating the distance from the point to the lidar; $\delta_{u}$ and $\delta_{v}$ are the angular resolution of the lidar’s rotation angle and pitch angle, respectively. Figure 2 shows a lidar image converted from a point cloud data. It provides a coordinate frame similar to the image to organize the discrete point cloud data, and it also maintains the spatial relationship between points. Considering that the driverless perception system requires high real-time performance of the algorithm, we expect to be able to operate directly on 2D images and avoid direct point cloud data computing in 3D space. Generally, for point clouds in the ground area and non-ground area, the horizontal angles of adjacent points in the same column are different in lidar images [23], and the height of these two types of points usually differs greatly. Based on these, this paper designs a discriminant image generation method. The formulas are as follows:(4)$$\Delta z=\left|d\left(u,v\right)sin\left(v\delta_{v}\right)-d\left(u-1,v\right)sin\left(\delta_{v}\left(v-1\right)\right)\right|;$$
(5)$$\Delta d=cos\left(v\delta_{v}\right)\left|d\left(u,v\right)cos\left(u\delta_{u}\right)-d\left(u-1,v\right)cos\left(\delta_{u}\left(u-1\right)\right)\right|.$$
(6)R(u,v)=arctan(Δz,Δd);
(7)$$G\left(u,v\right)=d\left(u,v\right)sin\left(u\delta_{u}\right)+H;$$
(8)B(u,v)=0.

Among them, R(u,v) is the R channel value of each point of the newly generated image and stores the value of the horizontal angle between the point and the adjacent point, as shown in Figure 3. G(u,v) is the G channel value of each point of the newly generated image that stores the absolute height value of the point, H  is the installation height of the lidar, and the B channel value is filled with zero values. Figure 4 shows the discriminant image converted from Figure 2. Figure 5 shows the expected lidar image after filtering out the ground points by using the GSDI algorithm.

We use breadth-first search (BFS) to traverse each column of the discriminant image,$\;M_{judge}$ to find all ground points. In order to filter out the pixels corresponding to the ground points in the image faster, we assume that the scanning line with the smallest pitch angle of the lidar is located on the ground. Therefore, the end element of each column in the image at the initial moment is marked as a ground point. The traverse algorithm flowchart is shown in Figure 6, and the detailed procedures are as follows.1.First, create a label image,$\;\;M_{label\;}$, equal to the size of $\;M_{judge\;}$ and initialize it as a zero-value matrix. At the same time, create a queue for storing the current ground point.2.Traverse $\;M_{\mathrm{judge}}$ from the first column of it and put the last element of the first column into queue.3.Take out the first element of the queue and mark it as a ground point at the corresponding position of $\;M_{\mathrm{label}\;}$. Judge the four neighbor points of this point. If the R channel value difference (angle difference) between the neighbor point and this point meets the threshold condition, its G channel value difference (height difference) is also within the threshold range. This neighbor point is also marked as the ground point, so store it at the end of queue. Otherwise, judge the next neighbor point until all neighbor points have been judged.4.Judge whether queue is null or not. If it is not null, then repeat Step 3. Otherwise, put the last element of the next column into the queue, and repeat Step 3, until all columns of $\;M_{\mathrm{judge}\;}$ have been traversed.5.According to the obtained label image, $\;M_{\mathrm{label}\;}$, the ground point is filtered out at the corresponding position on the lidar image, and then the lidar image without ground pixels is obtained, as shown in Figure 5. After obtaining the lidar image without ground pixels, point cloud can be projected into image, to obtain 3D point cloud without ground points.

### 2.2. Region of Interest

In this paper, the output 2D object detection frame is used to reduce the search space of the 3D object, which can be used as auxiliary information for the subsequent generation of the region of interest. The choice of the image detection method is flexible, as long as it can obtain the bounding box information of the object under the image coordinates, such as Mask RCNN [24], YOLO v3 [2], and so on. Through experimental comparison, we find that no matter in the detection accuracy and speed, YOLO v3 have reached the most advanced level of the single stage detection method, so YOLO v3 network is used as 2D detection module in our algorithm. The input of this 2D detection model are images, and the output are bounding boxes which include object’s position, object’s class, and confidence. After the installation position of the lidar and the visual sensor on the autonomous vehicle are measured (such as in Figure 7a [5]), the acquired point cloud data can be mapped into image through the spatial position relationship between the two. Then we can use the 2D detection box of the image to filter the 3D point cloud of the object.

Specifically, for a certain point, $P\left(X_{L},Y_{L},Z_{L}\right)$, we first need to convert it into the camera coordinate, $O_{C}$, and then project it into the image coordinate, o, and finally translate it to the position of the point under the pixel coordinates. The result is expressed as (u,v), and f  is the focal length of the camera, as shown in Figure 7b. Equation (9) shows the coordinate conversion formula.(9)$$\left[\begin{array}{c} u\\ v\\ 1 \end{array}\right]=\left[\begin{array}{ccc} \frac{f}{dx} & 0 & \begin{array}{cc} u_{0} & 0 \end{array}\\ 0 & \frac{f}{dy} & \begin{array}{cc} v_{0} & 0 \end{array}\\ 0 & 0 & \begin{array}{cc} 1 & 0 \end{array} \end{array}\right]\left[\begin{array}{cc} R & t\\ 0 & 1 \end{array}\right]\left[\begin{array}{c} X_{L}\\ Y_{L}\\ Z_{L} \end{array}\right].$$

All the point clouds after filtering out the ground are subjected to the above conversion, and the projection distribution of the point cloud data in the image can be obtained, as shown in Figure 8. It should be noted that this task only processes point cloud data that fall into the camera’s viewing angle, and points that are not within this range are not involved in the calculation. In the obtained projection image, the color of the dot matrix represents the depth of field; the warm tone represents distant points, and the cool tone represents close points. This paper marks all the suspected object points in the image frame area, then finds all the marked scanning points in the original point cloud data, and finally obtains the 3D area of the target object, as shown in Figure 9.

### 2.3. Dynamic Distance Threshold Clustering

In order to obtain the objects in the region of interest, we first created Kd-tree [25] for the data in the region, to organize the point cloud. Kd-tree is a commonly used point cloud data structure, which can effectively represent the set of points in 3D space. When searching for interval and neighbor points, the three dimensions of the point cloud are usually processed separately. However, because the detection result of the 2D image corresponds to only one target object, there is only one object in the region of interest theoretically, and there will be no stacks of objects in the Z direction. Therefore, the Kd-tree created in this paper is a two-dimensional Kd-tree. Point cloud object detection in a small range often uses Euclidean clustering. It uses the Euclidean distance between neighboring points as a criterion. When the distance value between two points is less than the given threshold, the points are considered to belong to the same cluster. However, the distance threshold of the traditional Euclidean clustering algorithm is always a constant value, so it is difficult to consider the clustering effect of the long-distance objects and the close-distance objects at the same time, resulting in the problem of over-segmentation of the long-distance objects. Based on this, this paper uses a method called DDTC to make the distance threshold, $T_{d}$, change with the distance of the point, as shown in Figure 10.

In this method, in order to represent the relationship between the current scanning point, $P_{i}$, and the threshold point, $P_{max}$, this paper defines a virtual line, L, that passes through the current scanning point, $P_{i}$ and assumes that its angle to the laser line is δ. Under this constraint, the current point, $P_{i},$ can be related to the threshold point, $P_{max}$, through the distance value from point O to the virtual line, L:(10)$$\left|OA\right|=sin\left(\delta \right)\sqrt{x_{i}^{2}+y_{i}^{2}}=sin\left(\delta -\Delta \alpha \right)\sqrt{x_{max}^{2}+y_{max}^{2}.}$$

Among them, $\left(x_{i},y_{i}\right)$ and $\left(x_{max},y_{max}\right)$ are the current scanning point, $P_{i}$, and the threshold point, $P_{max}$, in the XY plane, respectively, and Δα represents the horizontal angular resolution of the lidar. According to the principle of similar triangle, Equations (11)–(13) can be obtained.
(11)$$\frac{\left|BP_{max}\right|}{\left|OA\right|}=\frac{sin\left(\Delta \alpha \right)\left|OP_{max}\right|}{sin\left(\delta -\Delta \alpha \right)\left|OP_{max}\right|};$$
(12)$$\left|P_{i}P_{max}\right|=\frac{\left|BP_{max}\right|}{\left|OA\right|}\left|OP_{i}\right|;$$
(13)$$\left|OP_{i}\right|=\sqrt{x_{i}^{2}+y_{i}^{2}}.$$

Then, we can derive Equation (14).
(14)$$T_{d}\left(i\right)=\left|P_{i}P_{max}\right|+3\sigma_{r}=\frac{sin\left(\Delta \alpha \right)}{sin\left(\delta -\Delta \alpha \right)}\sqrt{x_{i}^{2}+y_{i}^{2}}+3\sigma_{r}.$$

In Equation (14), $\left|P_{i}P_{max}\right|$ is the Euclidean distance between the current scanning point, $P_{i}$, and the threshold point, $P_{max}$, in the XY plane, that is, the Euclidean clustering distance threshold of the $i^{th}$ scan point calculated in this paper, called $T_{d}\left(i\right)$. $\sigma_{r}$ represents the measurement error of the lidar. Through this calculation, we get a threshold circle with radius $T_{d}\left(i\right)$ and $P_{i}$ as the center, as shown by the blue circle in Figure 10. Obviously, the value of δ determines the final distance threshold. The larger the value of δ, the smaller the distance threshold. By selecting the appropriate δ according to the actual situation, the point’s Euclidean clustering threshold can be adjusted dynamically as the distance changes. In this paper, $\delta =10^{{}^{\circ}}$ is selected based on empirical judgment. After performing the DDTC method on all points of interest, we calculate the centroid of each point cloud cluster as the center of the object, and we calculate the cluster’s length, width, and height, to determine a 3D bounding box. Finally, 3D object detection results can be obtained, as shown in Figure 11.

## 3. Results and Discussion

### 3.1. Test Dataset and Evaluation Criteria

This paper uses the KITTI dataset to experiment and test the proposed algorithm. The KITTI dataset was jointly created by Karlsruhe Institute of Technology in Germany and Toyota American Institute of Technology. It is currently the world’s largest computer vision algorithm dataset for driverless scenes. The dataset is divided into two parts, a training set and a test set. The training set’s data are labeled and provides information such as the 3D size and position of the objects in the point cloud data. We randomly selected 7481 frames of training data, at a ratio of 4:1, and extracted 1500 frames of data as the verification set of this algorithm that is used to verify the performance of the algorithm. The evaluation criteria of the experiment follow the indicators proposed in the literature [16], that is, the average accuracy of the bird’s eye view box (AP loc) and the average accuracy of the 3D box (AP 3D). The larger the calculated AP value is, the better the detection performance will be. The operating software environment of the experiment is Ubuntu16.04 and ROS Kinetic. The computer hardware used is equipped with a 4.0 GHz Intel Core I7 processor and NVIDIA GeForce GTX 1080 graphics card.

### 3.2. Comparative Analysis of Experimental Results

The comparison methods of this experiment are mainly divided into three groups, according to different inputs, which represent use only image information for detection, use only lidar information for detection, and combine lidar information and image information to realize detection, respectively. These methods include Mono3D [6], Deep3DBox [7], 3DOP [8], and VeloFCN [19], which are based on image data; BirdNet [16], which is based on point cloud data; and F-PointNet, using image and point cloud data fusion [26]. In the experimental setting, this paper sets the intersection over union (IoU) threshold to 0.5 and 0.7. IoU is mainly used to measure the level of overlap between the ground truth box the bounding box generated by the model. We calculated the average accuracy value of the object detection box under these two different thresholds. The average accuracy value (AP loc) of the bird’s-eye view box is shown in Table 1. The average accuracy of the 3D detection box (AP 3D) is shown in Table 2, and the average time consumption of different algorithms is shown in Table 3.

As shown in the comparison results of Table 1 and Table 2, the AP loc and AP 3D of our algorithm and the point-cloud-based target detection method are significantly better than the image-based method only. This shows the superiority of lidar-based object detection in 3D perception. Compared with the method based on point cloud only, our algorithm is superior to BirdNet and VeloFCN based on point cloud projection. This shows that it is very helpful to use images as auxiliary information when searching for 3D objects. It can effectively improve the average accuracy of 3D object detection. When IoU = 0.7, the algorithm performs poorly and has a gap compared with F-PointNet. One possible reason is that F-PointNet uses category information of objects as prior information when estimating the box. Therefore, in future work, we will take this type of prior information into account and use it to modify the generated bounding box. It can be seen from Table 3 that the average time-consumption of the algorithm is the least among all methods, and the average time for processing one frame is only 74 ms. Among all the algorithms, BirdNet is more time-consuming than our algorithm, but its AP value is significantly lower. Although F-PointNet can achieve the highest average accuracy value, its run time is more than twice that of our algorithm. Generally, the scanning frequency of lidar is 10 Hz, that is, one frame of data will be acquired every 100 ms, which indicates that the running time of the target detection algorithm of 3D point cloud must be controlled within 100 ms. Thus, in summary, while maintaining a high degree of accuracy, the run time of our algorithm is also within the 100 ms time range required by lidar, which can meet the accuracy and real-time requirements of unmanned target detection.

## 4. Experimental Analysis of each Module of the Algorithm

This section analyzes the effectiveness of each module of our algorithm, and Section 4.1, Section 4.2 and Section 4.3 refer to module 1, module 2, and module 3 of our algorithm, respectively.

### 4.1. Ground Segmentation

In order to verify the performance and effectiveness of the ground segmentation algorithm, this section analyzes the algorithm for several typical scenes in autonomous driving and compares it with the mainstream method. Figure 12a shows the experimental results in the slope scene. The method based on occupancy grid map and our method has a better response to the slope scene. However, the ground in the slope scene is not a flat plane, and it is difficult for the usual plane model to fit it well. Therefore, there is a serious ground misdetection for the method based on ground plane fitting. Figure 12b shows the experimental result in the multi-obstacle scenes. Our algorithm can effectively distinguish between the obstacle point and the ground point, but the method based on occupancy grid map has the problem that the obstacle points in the suspended plane are mistakenly detected as ground points. Figure 12c is the experimental result in the scenes of multiple dangling objects. For the method based on occupancy grid map, the grid containing the dangling object points is judged as a non-ground-point grid, resulting in the problem of missing detection of ground points. Our algorithm and the method based on ground plane fitting perform better in this scene.

Table 4 gives quantitative statistics on the average time and run frequency of each algorithm. From the experimental results, we know that the run time of our algorithm is basically close to the method based on occupancy grid map, which effectively improves the run speed of the algorithm while ensuring the integrity of the point cloud data, and the processing frequency of 182 Hz fully meets the real-time requirements of unmanned driving.

### 4.2. Regions of Interest

An image detector is used to generate a region of interest when detecting 3D objects. This reduces the amount of calculation and obtains the suspected region of the object. In order to verify the performance of this method, we compared this method with the downsampling method. The experiment generates quantitative statistics on the AP loc value of the bird’s-eye view box with IoU = 0.5 at different distances. The results are shown in Figure 13. Our method has higher average AP loc values than ordinary downsampling methods at different distances. The AP loc value of the bird’s-eye view box of this method can reach more than 80% under the short distance and the middle distance. This is because our algorithm does not reduce the target point cloud but screens the point cloud purposefully, thus having a better detection effect for some objects with few points and distant objects.

In addition, the experiment generates quantitative statistics on the average time-consumption of the algorithm under two methods for a total of 1500 frames of point clouds, as shown in Table 5. It can be known from the obtained experimental data that the algorithm takes an average of 74 ms per frame. Although the downsampling method reduces the amount of data, it still operates on all point cloud data, which takes 628 ms. Therefore, the calculation speed of our method is very advantageous, especially for real-time tasks, such as autonomous driving. It can effectively improve the efficiency of the 3D object detection algorithm.

### 4.3. Dynamic Distance Threshold Euclidean Clustering

In order to show the improved performance of our algorithm compared to the traditional fixed distance threshold Euclidean clustering algorithm, we compared these two algorithms similarly, as shown in Figure 14 and Table 6. From the comparison results, we can see that, at a close distance, the AP loc value of the bird’s-eye view frame of the traditional Euclidean clustering method is almost the same as that of our algorithm, and the average detection accuracy within 10 m can reach 93.06%. However, as the distance increases, its AP loc value continues to decline. When the distance range of objects is 50~60 m, the traditional Euclidean clustering method only has an average accuracy value of 15.48%, while our algorithm in this paper still maintains an average accuracy value of 64.05%. The reason for this result is that the traditional Euclidean clustering algorithm with a fixed distance threshold has a probability of over-segmentation phenomenon for distant objects, as shown in Figure 15. Due to the adjustment of the distance threshold, our algorithm can well avoid this phenomenon, so the use of the dynamic distance threshold can effectively improve the detection accuracy of long-distance objects.

### 4.4. Visualized Results

In order to show the qualitative object detection results of this algorithm, this paper visualizes the results of the algorithm for the typical cases of unmanned actual road scenes, as shown in Figure 16a–f. Figure 16a shows that our algorithm can easily deal with small objects in the dark. Figure 16b shows that the algorithm can detect vehicles that are far away. Figure 16c shows that our algorithm also has a better detection effect on slightly occluded objects. Figure 16d is a typical multi-pedestrian scene, where our algorithm gives accurate detection results for the pedestrians present. Figure 16e illustrates that our algorithm can better cope with multi-vehicle scenes. Although there is a slight overlap, they can still be clearly separated. However, if the vehicle has a very serious occlusion, our algorithm will fail. For example, if most of the vehicles’ bodies were blocked during detection, bounding boxes cannot bound these cars, so they cannot be detected by 3D detection either. This problem can be improved by combining the detection results of the bird’s eye view, which is also a problem that this subject needs to solve in the future. Figure 16f shows that the algorithm has a good detection effect on different types of objects on the road.

## 5. Conclusions

Autonomous driving technology can fundamentally solve the traffic problems caused by human factors, and the lidar-based environment perception algorithm is a key component of the autonomous driving algorithm system, which is of great significance to the development of the autonomous driving field. In this paper, a fast object detection algorithm based on vehicle-mounted lidar was proposed. The algorithm proposes the GSDI method to convert point cloud data into discriminant images for threshold judgment, and ground points are filtered out efficiently. Then, the image detector is used to generate the region of interest of the 3D object, effectively narrowing the search range of the target. Finally, in view of the characteristic that the difference of the point cloud data distance will cause the density of the point cloud data to be different on the target surface, a DDTC method which uses dynamic distance threshold for the Euclidean clustering algorithm is designed, which effectively improves the detection accuracy of long-distance objects. By comparing with the mainstream 3D object detection algorithm, it was also shown that the algorithm can maintain high accuracy and also meet the real-time requirements of unmanned driving. Although the algorithm has a good detection effect in most scenes, the detection effect of obstacles under strong occlusion is not good. This problem can be improved by combining the bird’s-eye view detection results or by inferring the position of the obscured object in the current frame based on the past information of the object, which will be completed in the next research. It is worth mentioning that, although only the front-view camera was used in the study, if the vehicle is equipped with panoramic camera or multiple cameras covered 360-degree view, point cloud of 360 degree view of lidar also can be used. In actual application, if only one camera with a non-big FOV is applied, suppressing point clouds at the beginning will be very effective and will reduce the time consumption further.

## Figures and Tables

**Figure 1 sensors-20-07221-f001:**
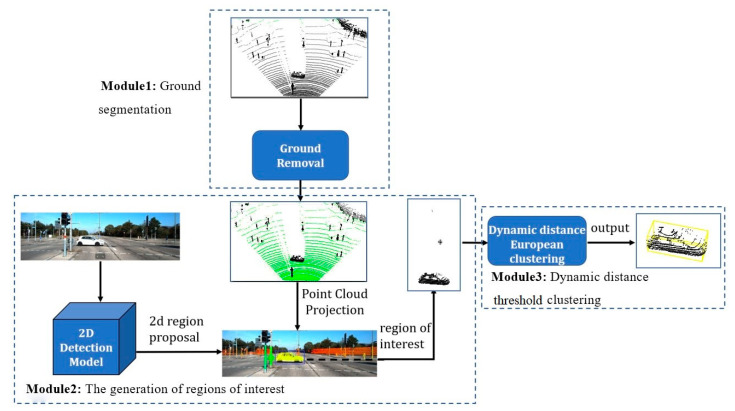
Algorithm framework.

**Figure 2 sensors-20-07221-f002:**
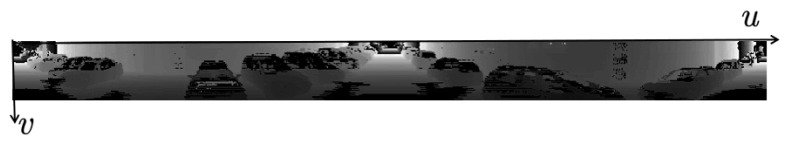
Lidar image.

**Figure 3 sensors-20-07221-f003:**
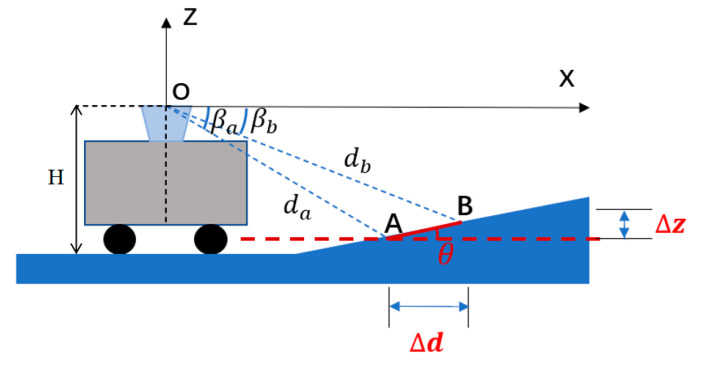
Horizontal angle of point.

**Figure 4 sensors-20-07221-f004:**
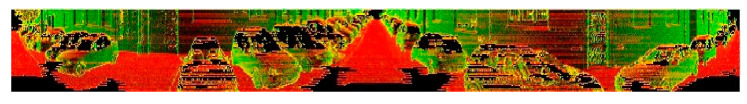
Discriminant image.

**Figure 5 sensors-20-07221-f005:**
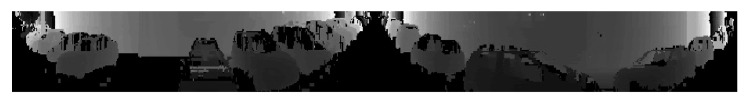
Lidar image after filtering out the ground.

**Figure 6 sensors-20-07221-f006:**
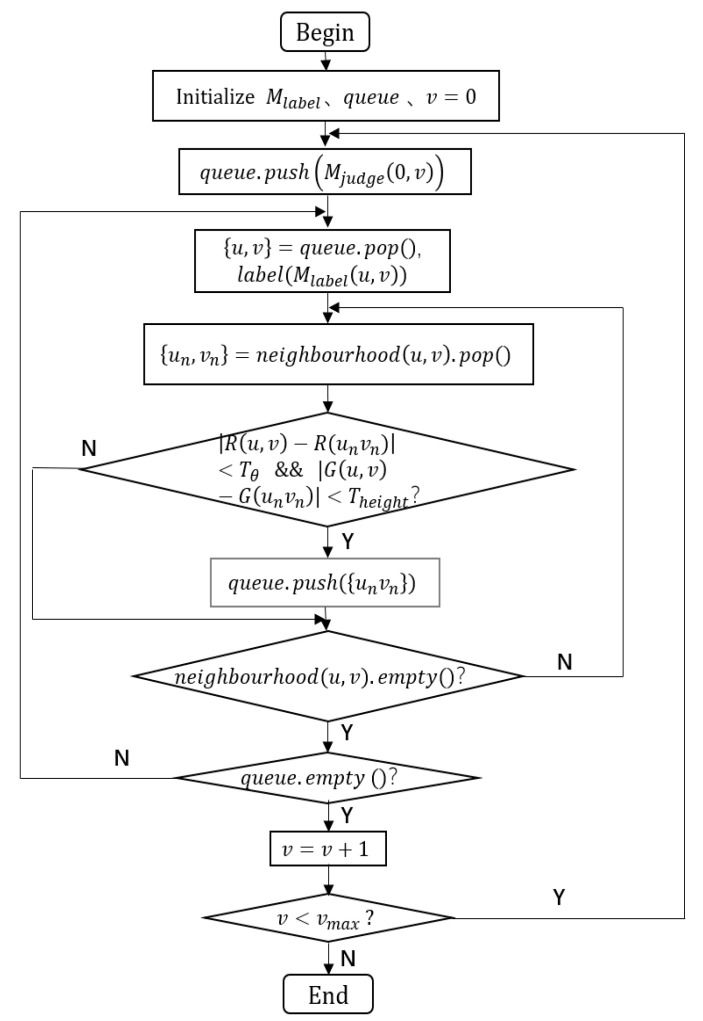
Flowchart of the traverse algorithm.

**Figure 7 sensors-20-07221-f007:**
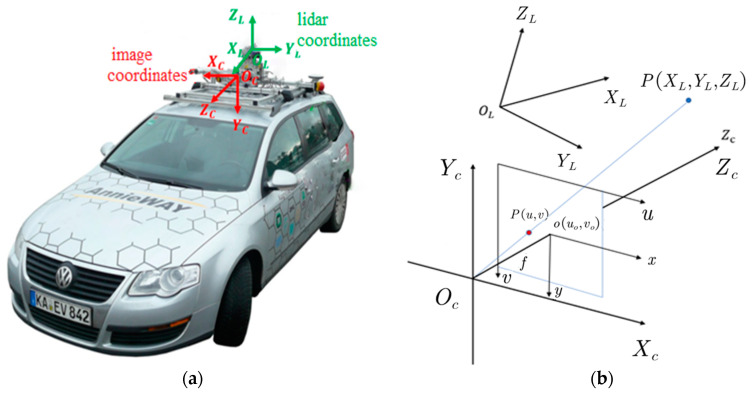
Conversion relationship between lidar coordinates and image coordinates. (**a**) shows an example of lidar coordinates and camera coordinates on a real vehicle. (**b**) shows how the lidar point cloud data is projected onto the image plane.

**Figure 8 sensors-20-07221-f008:**
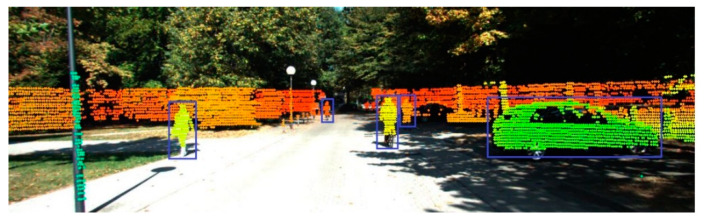
Point cloud projection results. The color of the dot matrix represents the depth of field, the warm tone represents distant points, and the cool tone represents close points. Suspected object regions are marked with blue bounding boxes.

**Figure 9 sensors-20-07221-f009:**
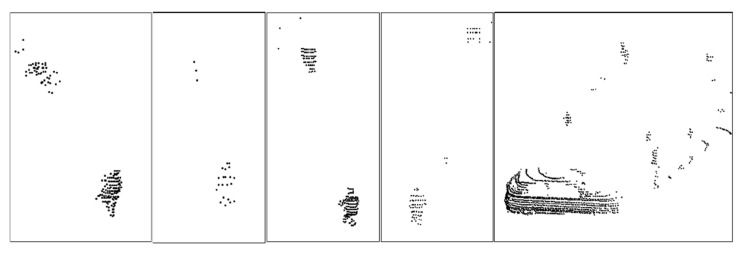
Regions of interest. It shows all the suspected object points in the image frame area, and we can obtain the real 3D area of the target object from it.

**Figure 10 sensors-20-07221-f010:**
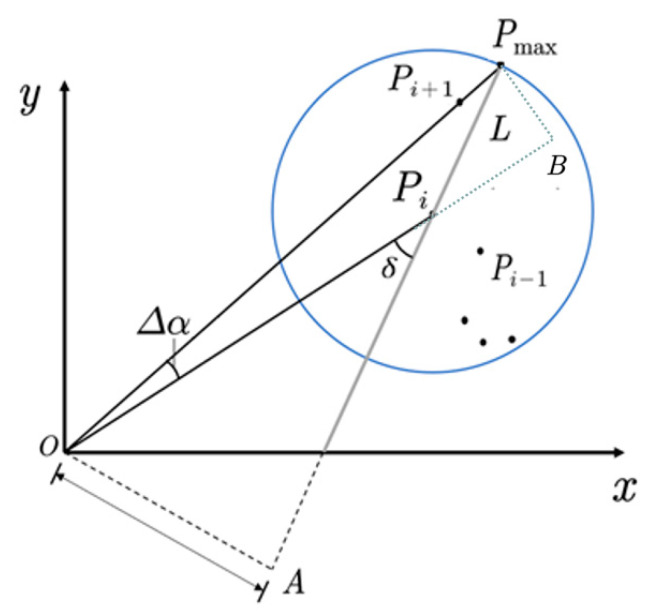
Calculation of distance threshold.

**Figure 11 sensors-20-07221-f011:**
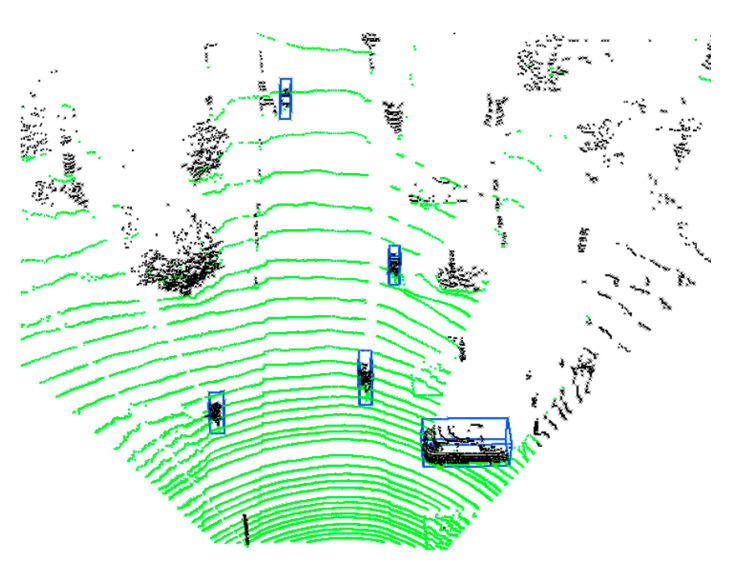
The final objects-detection results from the original point cloud are marked with blue bounding boxes.

**Figure 12 sensors-20-07221-f012:**
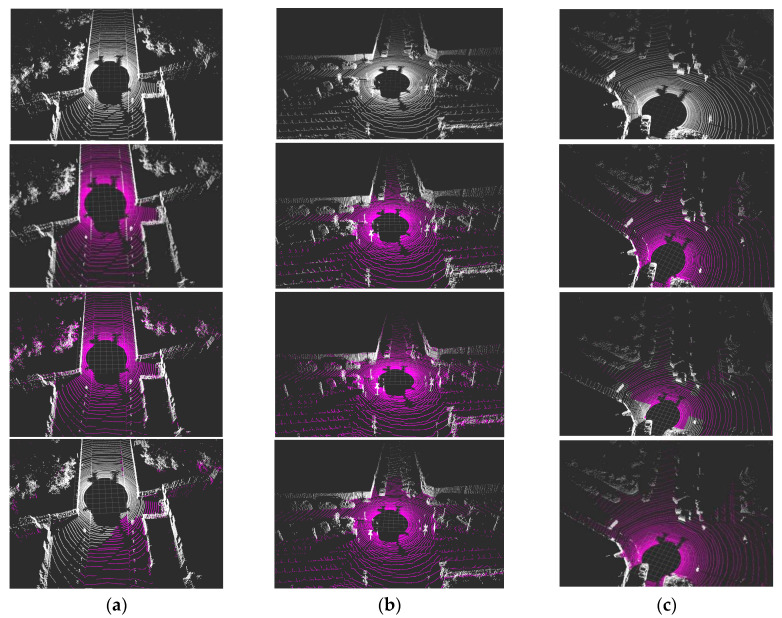
Experimental results of different ground segmentation algorithms: (**a**) detection result in slope scene, (**b**) detection result in multi-obstacle scene, and (**c**) detection result in multiple dangling objects scene. For each of these subfigures, the pictures from top to bottom are the original scene, the detection results of this method, the detection results of the method based on occupancy grid map, and the detection results of the method based on ground plane fitting, respectively.

**Figure 13 sensors-20-07221-f013:**
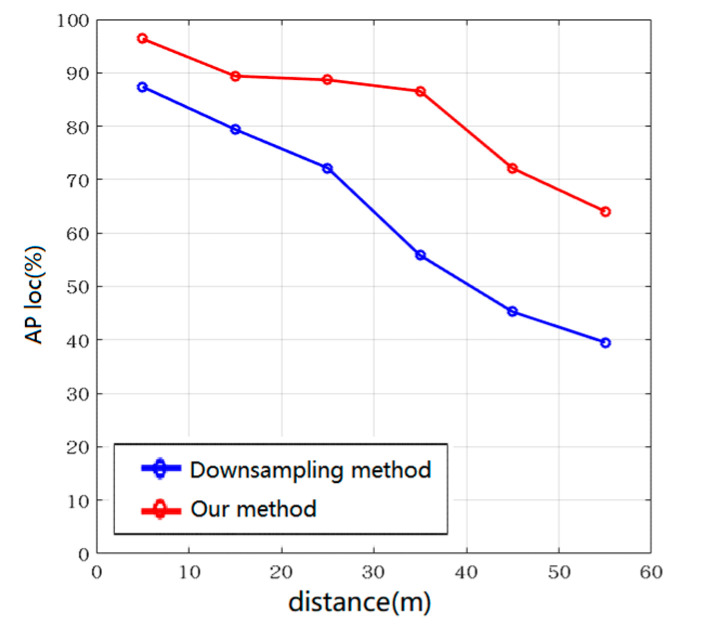
AP loc value at different distances.

**Figure 14 sensors-20-07221-f014:**
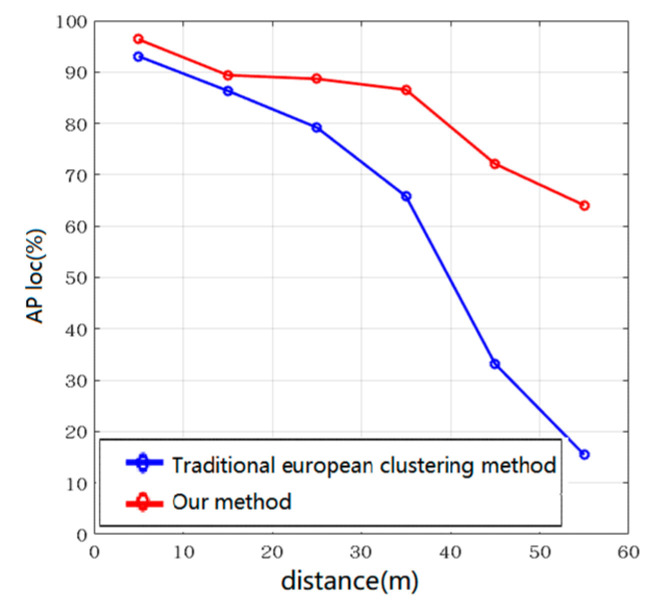
AP loc value at different distances.

**Figure 15 sensors-20-07221-f015:**
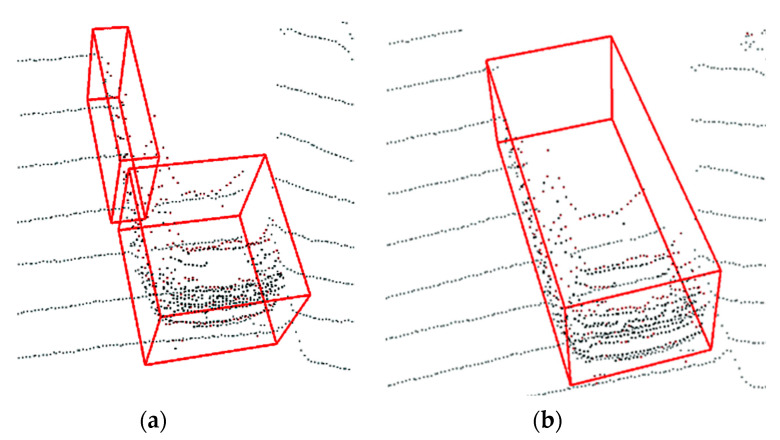
Comparison of long-distance object detection results: (**a**) over-segmentation phenomenon of the traditional Euclidean clustering method and (**b**) clustering result of the dynamic distance threshold Euclidean clustering method.

**Figure 16 sensors-20-07221-f016:**
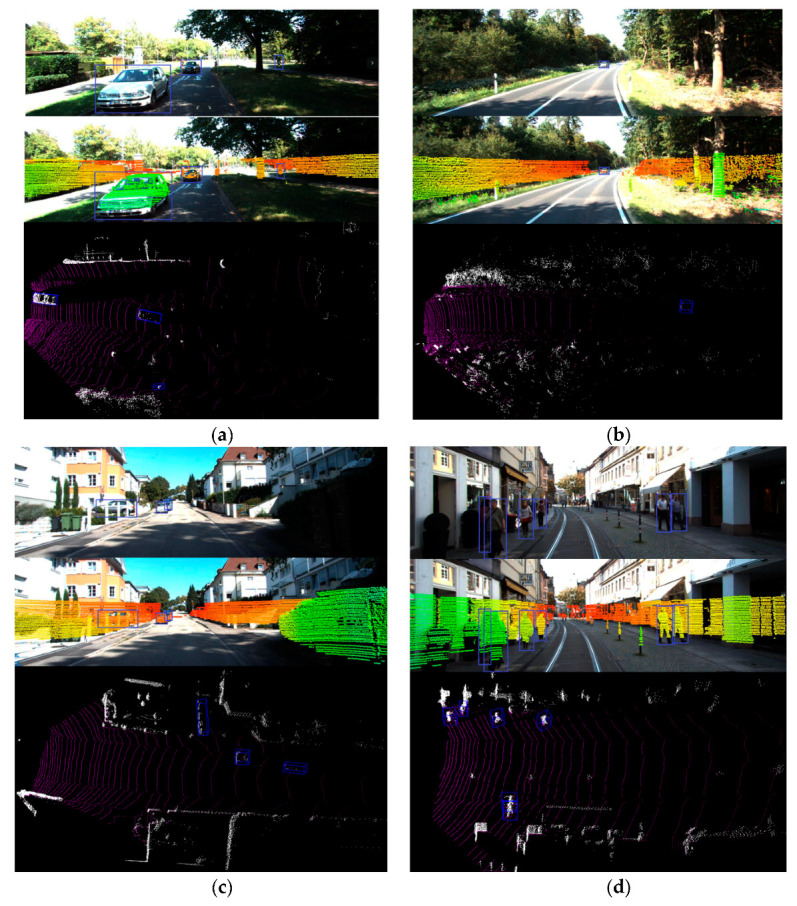
Visualization results under typical cases: (**a**) shows the detection result of objects in the dark, (**b**) shows the detection result of objects that are far away, (**c**) shows the detection result of objects that are slightly occluded, (**d**) shows the detection result of a multi-pedestrian scene, (**e**) shows the detection result of a multi-vehicle scene, and (**f**) shows the detection result of different types of objects on the road. For each subfigure, the top, central, and bottom pictures represent the detection result of 2D image detector, projection result of point cloud to image after filtering ground, and the final detection result, respectively.

**Table 1 sensors-20-07221-t001:** The average accuracy value of the bird’s-eye view box (AP loc).

Algorithms	AP loc (%)					
	IoU = 0.5			IoU = 0.7		
	Easy	Moderate	Hard	Easy	Moderate	Hard
Mono3D	30.50	22.39	19.16	5.22	5.19	4.13
Deep3DBox	29.96	24.91	19.46	9.01	7.94	6.57
3DOP	55.04	41.25	34.55	12.63	9.49	7.59
BirdNet	N/A	N/A	N/A	35.52	30.81	30.00
VeloFCN	79.68	63.82	62.80	40.14	32.08	30.47
F-PointNet	88.70	84.00	75.33	50.22	58.09	47.20
Our method	83.23	71.74	70.28	49.45	43.65	40.39

**Table 2 sensors-20-07221-t002:** The average accuracy of the 3D detection box (AP 3D).

Algorithms	AP 3D (%)					
	IoU = 0.5			IoU = 0.7		
	Easy	Moderate	Hard	Easy	Moderate	Hard
Mono3D	25.19	18.20	15.52	2.53	2.31	2.31
Deep3DBox	24.76	21.95	16.87	5.40	5.66	3.97
3DOP	46.04	34.63	30.09	6.55	5.07	4.10
BirdNet	N/A	N/A	N/A	14.75	13.44	12.04
VeloFCN	67.92	57.57	52.56	15.20	13.66	15.98
F-PointNet	81.20	70.39	62.19	51.21	44.89	40.23
Our method	74.68	63.81	60.12	32.79	29.64	21.82

**Table 3 sensors-20-07221-t003:** Average time-consuming.

**Algorithms**	Mono3D	Deep3DBox	3DOP	BirdNet	VeloFCN	F-PointNet	Our method
**Time (ms)**	206	213	378	110	1000	170	74

**Table 4 sensors-20-07221-t004:** Comparison of runtime.

Algorithms	Runtime	Frequency
Grid map-based	3.7 ms ± 0.2 ms	270 Hz
Ground plane fitting-based	24.6 ms ± 1.2 ms	41 Hz
Our method	5.5 ms ± 0.3 ms	182 Hz

**Table 5 sensors-20-07221-t005:** Average runtime.

Method	Average Runtime
downsampling	628 ms
region of interest	74 ms

**Table 6 sensors-20-07221-t006:** Comparison results of the two clustering methods.

Distance	Number of Actual Objects	Traditional Euclidean Clustering Algorithm		Our Method	
		Number of Objects Detected Accurately	AP loc	Number of Objects Detected Accurately	AP loc
0~10 m	3128	2911	93.06	3014	96.35
10~20 m	2952	2549	86.35	2639	89.39
20~30 m	2577	2041	79.20	2285	88.67
30~40 m	2705	1780	65.80	2341	86.55
40~50 m	2384	791	33.18	1719	72.12
50~60 m	2048	440	15.48	1312	64.05

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
