# Peer review of "3D Fast Object Detection Based on Discriminant Images and Dynamic Distance Threshold Clustering"

_sensors, 2020, doi:10.3390/s20247221_

Round 1
Reviewer 1 Report
This paper presents an object detection algorithm that attempts to achieve good object recognition properties but at an increased execution speed to comparable algorithms from the literature. The algorithm consists of three modules that interact to produce the desired outcome. The paper provides a very good survey of current recognition algorithms used in autonomous vehicle object recognition tasks using the KITTI 3D Object Suite and outlines the various problems associated with 3D object recognition.
The authors present a clear explanation of their algorithm and how the various components work together to provide object recognition/detection.
The results show that the algorithm is indeed very fast compared to others that are used as standard benchmarks and that the algorithm is of high accuracy. The results are presented in a clear format and show that the author’s claims are met.
It seems that further work on developing and refining the algorithm is worthwhile. I would also suggest that parallelizing the algorithm would help in speeding it up as well as providing multiple data sets for comparison during object detection. The use of neural network in detecting obscured items.
Suggested changes
Lines 317 and 324 replace “makes” with “generates”
Line 386 replace “proven” with “shown”
Reviewer 2 Report
This paper proposed a method for fast 3d object detection. GSDI to remove the ground was proposed, and the ground was removed. In addition, after extracting the point by specifying the ROI from the 2D image, the final object point was clustered by suggesting DDTC. As a result, high performance was maintained and a very fast algorithm was developed.
However, although two methods(GSDI, DDTC) were proposed, explanation of both methods is insufficient. Also, the experimentation to support the claim is weak. Author claimed that the high performance was maintained and the speed was fast. However, most of the compared algorithms rank low in the kitti competition. It should be compared to a higher ranking algorithm.
- Most figures have low resolution.
- Mark the u and v directions in Figure 3.
- Eq. (3) and (4) seem to be equations for calculating the height. Why did you use δv in Equation 3 and δu in Equation 4? then, is Equation 5 correct?
- Mark H in Figure 2.
- In Eq (8), the B channel has a value of 0, but it seems unnecessary.
- The description of the algorithm(Fig. 6) in GSDI is lacking. In particular, the angle of the floor was used as a feature, but there is insufficient explanation for that part.
- There is no clear reason for why it was proposed as in Eq. (11). Why did you use δ, and what is meaning of OA?
- Eq. (11) is the distance threshold equation. So, you need to add (td=)
- Author claims that the cluster results for long-distance objects became also very good. However, Figure 15 shows only the clustering results of nearby objects. Therefore, clustering comparison results of far objects should be added.
- In the result of Figure 16, the bounding box includes the size and orientation of the object. However, in this paper, there is no explanation on how bounding boxes were created.
Reviewer 3 Report
>>> Summary
The paper concentrates on autonomous driving and more precisely on the use of on-board lidars to perform 3-dimensional fast object detection in real-time. In this framework, the authors propose a strategy based on ground segmentation, generation of the region of interests, and dynamic clustering.
The first phase, ground segmentation, is based on a set of geometrical formulas that convert the lidar image into a set of 3-dimensional points.
The second phase, the generation of the region of interests, is heavily based on YOLO V3 to detect 2-dimensional objects and to create bounding-boxes, and then on the objects' coordinates conversion process.
The third phase, the dynamic clustering, is based on a 2-dimensional variation of the KD-tree algorithm (reference [25]) in which the distance threshold varies with the distance of the point.
Experimental results concentrate on the KITTI benchmark suite. The authors report a comparison with a few related strategies, image-based, cloud point-based, and image plus cloud point-based. These results show that the proposed tool achieves smaller manipulation times (down to 74ms per frame) and competitive results in terms of 2D and 3D precision.
>>> Strong Points
The paper presents interesting results which make the method valuable.
Note that results are convincing but not striking as existing tools beat the proposed framework at least in terms of precision.
>>> Weak Points
Unfortunately, the paper is also sloppily organized. Several sections and paragraphs should be reformulated. English style must be improved. Figures are often unreadable or misplaced.
Here there is a partial list of the main structural problems:
- The authors tend to enumerate the steps and then to describe them. An example of this style is at lines 14-15 then repeated at lines 16-23. The same consideration holds for lines 102-103 then repeated on lines 103-113. I think that the first numbering is superfluous as the second one rightly follows and it should be sufficient. Thus, the authors should try to merge these two sections in all cases.
- From line 41 to line 89, there is an introduction on point cloud detection methods, as they were in the early days and in the deep-learning era, respectively. I think these paragraphs should be postponed in a “related work” section appearing just after the introduction. Moreover, the description of these methods should follow the point cloud definition appearing at lines 89-91.
- Lines 103-113 describe the method in the introduction section. Unfortunately, these lines are pretty similar to the ones used to describe the method right in the abstract section (line 15-22). The content of this introductory part should be improved, made more detailed, and longer.
- Figure 1 (as Figures 6, 7b, 10, 13, 14, and 15) is quite unreadable. The authors should improve the quality of their figures and enlarge both the fonts and the pictures.
- In Figure 3 and others, the same figure spans more than one page or the caption is on the following page. This decreases the readability of the paper.
- The paper is missing a proper roadmap. Authors should at least say at the beginning of Section 2 (and of Section 4) that sub-sections 2.1, 2.2., and 2.3 (and 4.1, 4.2, and 4.3) refer to modules 1, 2, and 3 of their procedure, respectively.
- In the equation from 1 to 8, authors should try to use the same equation style (and fonts).
- The description of lines 257-259 (image-based, cloud-based, and image+cloud-based tools) should be highlighted with more accuracy, as the interpretation of the results strongly depends on which methods each tool do use to perform the same task.
- The authors should better describe the parameter IoU, as it has a very strong influence on the data of Table 1 and 2.
- The authors should clarify the meaning of the top, center, and bottom pictures for each one of the figures in Figure 16.
Some example of poor English usage follows:
- Line 12: “which is difficult to realize”, please rephrase.
- Line 22: “Experiments have verified …”, showed?
- Line 35: “due to the lack of depth data …”, the authors are referring to the lack of information related to the object depth; please rephrase
- Line 89: “different from the image, it …”, maybe “different from the image as it …”.
- Line 91: “It brings … of targets.”, Hug?
- Line 189 (and again 194): “distant … and close”; hug? Maybe “distant and close points/objects”?
>>> Minor Points
- YOLO V3 sometimes is YOLO v3.
- Line 356: Figure 16(a) to 16(f) not (e).
Reviewer 4 Report
The paper proposes a liDAR-based real-time object detection, the performance of the proposed method is verified using KITTI dataset. The paper is very well-organized and well written
In Section 2.2 line 186 you mentioned that this task only processes point cloud that falls into the camera viewing angle
Does this mean that the developed object detection technique can operate within the camera view only and not 360-degree view of the LiDAR? If yes why didn't you suppress the point clouds that are not in the camera view area from the beginning?
You mentioned that 1500 frames from KITTI data are used for evaluating the performance of the proposed method, Did you choose an urban area, highway, what are the lighting conditions (day/night)?? Do you think you will have limitations in low-light conditions?
The sentence on line 172 that starts with "Through experimental" needs to be rephrased. Also, did you do these experiments, if not citations are needed.
All the figures' resolution and size should be enhanced and the text within figures is barely readable.
Round 2
Reviewer 2 Report
- It seems Equation 11 is derived from Equation 10. It would be better to explain this more clearly.
- In Figure 15, showing the results by distance will be better to support your arguments. (ex 10m, 30m, 50m)
- The 3d bounding box is used in the experiment result. But the way of creating a 3d bounding box is not described. How did you make the 3d bounding box?
